# On Coolant Flow Rate-Cutting Speed Trade-Off for Sustainability in Cryogenic Milling of Ti–6Al–4V

**DOI:** 10.3390/ma14123429

**Published:** 2021-06-21

**Authors:** Asif Iqbal, Guolong Zhao, Hazwani Suhaimi, Malik Muhammad Nauman, Ning He, Juliana Zaini, Wei Zhao

**Affiliations:** 1Faculty of Integrated Technologies, Universiti Brunei Darussalam, Jalan Tungku Link, Gadong BE 1410, Brunei; hazwani.suhaimi@ubd.edu.bn (H.S.); malik.nauman@ubd.edu.bn (M.M.N.); juliana.zaini@ubd.edu.bn (J.Z.); 2Department of Manufacturing & Automation, College of Mechanical & Electrical Engineering, Nanjing University of Aeronautics and Astronautics, 29-Yu Dao Street, Nanjing 210016, China; zhaogl@nuaa.edu.cn (G.Z.); drnhe@nuaa.edu.cn (N.H.); nuaazw@nuaa.edu.cn (W.Z.)

**Keywords:** machining, titanium, tool damage, surface roughness, tungsten carbide, cutting energy, material removal rate, liquid nitrogen, CO_2_ gas, cutting forces

## Abstract

Application of cryogenic fluids for efficient heat dissipation is gradually becoming part and parcel of titanium machining. Not much research is done to establish the minimum quantity of a cryogenic fluid required to sustain a machining process with respect to a given material removal rate. This article presents an experimental investigation for quantifying the sustainability of milling a commonly used titanium alloy (Ti–6Al–4V) by varying mass flow rates of two kinds of cryogenic coolants at various levels of cutting speed. The three cooling options tested are dry (no coolant), evaporative cryogenic coolant (liquid nitrogen), and throttle cryogenic coolant (compressed carbon dioxide gas). The milling sustainability is quantified in terms of the following metrics: tool damage, fluid cost, specific cutting energy, work surface roughness, and productivity. Dry milling carried out the at the highest level of cutting speed yielded the worst results regarding tool damage and surface roughness. Likewise, the evaporative coolant applied with the highest flow rate and at the lowest cutting speed was the worst performer with respect to energy consumption. From a holistic perspective, the throttle cryogenic coolant applied at the highest levels of mass flow rate and cutting speed stood out to be the most sustainable option.

## 1. Introduction

Titanium alloys are among the most important engineering alloys, which find widespread applications in various engineering sectors. Machining is an inevitable shaping requirement through which the alloys have to pass for their structural applications. Ti–6Al–4V, the most commonly used titanium alloy, is cut with poor machinability [1]. A high rate of heat generation is due to the alloy’s high shear strength, and the accumulation of the heat close to the cutting edge is due to a shorter-than-normal chip-rake interface cause intensification of heat flux around the cutting zones [2]. The intense heat flux accelerates the temperature-dependent modes of tool damage and thus seriously shortens tool life and dents machining sustainability. Therefore, it becomes a prime requirement to efficiently dissipate the accumulated heat through the application of an effective coolant.

Machining of Ti–6Al–4V under a cryogenic coolant has been thoroughly investigated and its benefits in improving machinability are well established. However, an improvement in machinability—mostly quantified in terms of rate of tool wear and tool life—does not necessarily mean an enhancement in sustainability, as the latter depends on other metrics as well, such as energy consumption, waste generation, process cost, etc. [3]. A vast majority of published work that has focused on studying the effects of cryogenic cooling in titanium machining has done so without considering the effects of varying the coolant’s mass flow rate. Machining sustainability does not seek just the optimization of the coolant’s flow rate but also its adaptation with respect to the material removal rate. This inquest sits at the heart of this work. In other words, mass flow rate of a cryogenic coolant per unit material removal rate needs to be optimized for maximization of machining sustainability. In this context, the forthcoming paragraphs provide a brief review of literature published with respect to titanium machining, application of cryogenic coolants, effects of coolant’s mass flow rate and cutting parameters, and evaluation of sustainability measures.

Due to poor machinability of the titanium alloys, it is quite difficult to exceed the cutting speed of 60 m/min using conventional tooling [4]. Advanced tool materials, such as ceramics, diamond, and cubic boron nitride are therefore used for enhancement in the productivity levels. Yip et al. presented a quantification approach for the evaluation of triple bottom line of machining sustainability in ultraprecision machining of titanium alloys [5]. Machining sustainability is generally quantified in terms of tool life, specific energy consumption, processing cost, material removal rate, work surface quality, and amount of swarf generated [3,6]. Regarding the milling of Ti–6Al–4V, the effect of cutting speed on temperature is more prominent than those of feed rate and depth of cut [7]. This suggests that the practitioners are bound to keep a check on cutting speed to keep the magnitudes of the temperature-dependent tool wear modes in their acceptable ranges.

Cryogenic cooling has been extensively explored in last 25 years for improving the machinability of titanium alloys. An experimental study has concluded that the performance of carbide tools is better than cubic boron nitride in the continuous turning of Ti–6Al–4V, conducted at high cutting speeds and under high pressure of the coolant’s supply [8]. It was reported that with a metered and properly directed application of liquid nitrogen (LN_2_), the tool life can be increased to five times of that yielded by a standard emulsion-based coolant [9]. While machining under LN_2_, the hot hardness of the tool remains high, whereas the temperature-dependent tool modes are significantly quashed [10]. Ayed et al. investigated the effects of varying spray-nozzle diameter on tool wear and work surface integrity in the cryogenic machining of a titanium alloy (Ti–6Al–4V) [11]. The increase in flow rate and pressure of the cryogenic fluid had a positive impact on the tool life, whereas the best surface integrity was found at the highest levels of jet pressure and flow rate. Tool life was reported to have significantly improved as the flow rate of the cryogenic coolant (liquid CO_2_) is increased [12]. The effect of the cryogenic fluid was found to have surpassed that of an emulsion-based conventional coolant at all the flow rates. One study explored the effects of supplying the cryogenic fluid (LN_2_) through duplex jets, aiming at the rake and flank faces [13]. The approach was found to have prolonged tool life by 60% and 30% in comparison with dry cutting and single-jet cryogenic cutting, respectively. Likewise, CO_2_ snow was reported to be used as a cryogenic coolant at different locations of a grooving tool with and without the merger of microlubrication in continuous machining of a high-strength β-titanium alloy [14]. The results showed that the hybridization of the cryogenic fluid and the location of its application have significant effects on the sustainability measures. CFD-based analyses and experimental verifications were performed to assess the effects of the coolant’s pressure in the cryogenic milling of Ti–6Al–4V [15]. Application of LN_2_ at low pressures (0.2–0.4 MPa) showed favorable results regarding cooling efficiency, the coolant’s consumption rate, and flow transfer efficiency. Likewise, another investigation showed that the cooling effectiveness of LN_2_ jet depends a lot on jet pressure, jet velocity, and pipe geometry [16]. An experimental investigation compared the sustainability of applying LN_2_ in an innovative “between-the-holes” mode with the conventional “continuous supply” mode in the drilling of Ti–6Al–4V [17]. The former was found to yield better surface quality and lower energy consumption but could not perform well in terms of tool life or productivity. The effect of cryogenic cooling using LN_2_ was also investigated in high-speed machining of gamma titanium aluminides [18]. It was reported that the impact of the cryogenic fluid was highly effective in counteracting the intense thermal load around the cutting edges.

The literature reviewed above suggests that although LN_2_, as a cryogenic coolant, possesses immense potential to improve the machinability of titanium alloys, its supply has not been optimized for the highest levels of sustainability. A research gap needs to be filled regarding the effects of trade-off between the coolant’s flow rate and material removal rate on the machining’s sustainability measures. In other words, the coolant’s flow needs to be metered with respect to the levels of cutting parameters employed in a machining process. In this context, the article presents an experimental investigation to adjust the cryogenic coolant’s mass flow rate with respect to cutting speed in side- and end-milling of Ti–6Al–4V.

## 2. Materials and Methods

The section provides the details on the tooling and work material, the controlled and measured parameters, the experimental setup, the design of experiments, and the measurements and the relevant instruments.

### 2.1. Work Material and Milling Tools

The study focuses on the cutting of an α + β alloy of titanium, Ti–6Al–4V. The dimensions of the solid blocks used in the experiments are 200 mm × 100 mm × 20 mm. The annealed temper of the alloy was obtained by heating the blocks around 780 °C for about 75 min, followed by cooling in open air. The post-annealed work material achieved the following values of the important mechanical properties: yield strength (at 0.2% proof stress), 930 MPa; ultimate tensile strength, 1005.7 MPa; and elongation, 14%.

The milling tools used in the experiments are TiN and TiAlN coated tungsten carbide flat end mill cutters. The cutters possess a helix angle, number of flutes, fluted length, and total length of 42°, 4, 63 mm, and 20 mm, respectively. The surface hardness and coefficient of friction in the fluted area are 3300 HV and 0.6, respectively.

### 2.2. Controlled and Measured Parameters

The following parameters were controlled in the experiments:Coolant: The three levels of this predictor were dry (no coolant at all), evaporative cryogenic coolant (LN_2_), and throttle cryogenic coolant (compressed CO_2_ gas).Coolant’s mass flow rate (kg/min): The three levels controlled were 0.2, 0.4, and 0.6 kg/min.Cutting Speed, *V*_c_, (m/min): The four levels controlled were 50, 100, 150, and 200 m/min.

A full-factorial design of experiments generated a total of 28 (=1 × 4 + 3 × 4 + 3 × 4) runs. It is to be noted that the first parameter is categorical whereas the other two are numeric. The four levels of cutting speed are so selected to comprehensively cover the feasible range of titanium machining. Moreover, most of the cutting speed selections, as reported in the published work related to titanium milling, fall in the 50–200 m/min range. On the other hand, the range for mass flow rate (0.2–0.6 kg/min) was selected based on the results of screening tests for both types of the cryogenic coolants. Anything less than 0.2 kg/min could not yield adequate cooling even for the lowest cutting speed, whereas the flow rate of more than 0.6 kg/min caused turbulence in the coolant’s flow, especially for the throttling CO_2_ gas. The following parameters (responses) were measured/evaluated for each experimental run:Average width of flank wear land, *VB* (mm), as observed on the four flutes of the used tool, to be measured after completing the run.Specific cutting energy, *SCE* (J/mm^3^), averaged for the entire cut length of each run.Average arithmetic roughness of the machined surface, *R*_a_ (µm).Machining forces, *F*_xy_ (N) and *F*_z_ (N), averaged for the entire length of cut. The former is the resultant of the two coplanar components *F*_x_ and *F*_y_, whereas the latter is the orthogonal component directed along the axis of the milling cutter.Specific fluid consumption, *SFC* (kg/m). For each run, the *SFC* is obtained by dividing the given mass flow rate of a coolant with the tool’s feed speed.

### 2.3. Experimental Setup

Figure 1 presents the experimental setup. All the experimental runs were performed on a CNC 5-axis vertical machining center that possesses a maximum rotational speed, feed speed, and main motor power of 18,000 rpm, 20 m/min, and 16 kW, respectively. The milling runs were performed in a linear direction, cutting through the 100 mm long side with the radial and axial depths of cut of 0.5 and 7 mm, respectively. Each experimental run consisted of two passes of the cut, thus removing a total of 700 mm^3^ (2 passes × 100 mm × 0.5 mm × 7 mm) of work material’s volume. The feed per tooth (*f*_z_) was fixed to 0.1 mm/z. Under this feed rate, the four levels each of feed speed (*V*_f_) and material removal rate (*MRR*), obtained against the four levels of cutting speed, are presented in Table 1. A new cutter was used for each run and was so held in the collet to have a protrusion length of 30 mm. The milling orientation used for all the runs was down-milling.

An evaporative cryogenic coolant is a supercool liquid that absorbs heat from a hot region and evaporates into the atmosphere. The amount of heat absorbed is equal to the mass of the evaporated coolant times its enthalpy of vaporization. This work utilized LN_2_ as the evaporative coolant, operating at a temperature of about −196 °C. The LN_2_ was transported from a storage dewar to the milling area through a thermally insulated pipe fitted with a nozzle of 6 mm diameter. The regulating valve was carefully controlled to obtain the volumetric flow rates of 0.248, 0.495, and 0.743 L/min, which correspond, respectively, to the mass flow rates of 0.2, 0.4, and 0.6 kg/min against the coolant’s specific gravity of 0.808 at −196 °C. The acquisition cost of the evaporative cryogenic coolant was 12 CNY/kg (as on 15 February 2021 in Nanjing, China).

A throttle cryogenic coolant is a compressible fluid that produces a cooling effect in the vicinity when it expands through an orifice. The phenomenon is known as Joule–Thomson effect [19]. Compressed CO_2_ gas was used as the throttle cryogenic coolant in this work. The throttling CO_2_ gas was found to attain a temperature of as low as −72 °C, whereby it deposits on an exposed surface in form of a quasi-solid structure, known as snow [20]. Thereafter, the deposited snow gradually sublimated into the atmosphere. The throttle coolant was transported from a storage bottle, where it was maintained under a pressure of 5.5 MPa, to the milling area through a tube fitted with a nozzle of 2 mm diameter. The nozzle was so directed to have impingement of the throttling fluid directly on that region of the milling cutter which is intermittently engaged with the work. The storage bottle’s regulating valve was carefully controlled to have the fluid’s mass flow rates of 0.2, 0.4, and 0.6 kg/min. The acquisition cost of the throttle cryogenic coolant was 3 CNY/kg (as on 15 February 2021 in Nanjing, China).

The supply of the relevant cryogenic coolant was opened 20 s prior to the first engagement of the tool with the work surface and was stopped immediately at the completion of the run.

### 2.4. Measurements

A Mahr surface roughness meter, the MarSurf M 300 C, was used to measure the arithmetic roughness of the side surface generated by the milling process. Four measurements were taken at the distances of 20, 40, 60, and 80 mm from the starting point. The sampling length for each measurement was 4 mm. The *R*_a_ for each run was then evaluated by taking the average of the four measurements. An optical microscope was used to measure the *VB* of the used milling cutters. Images of the flank wear land that formed on the four teeth, at the end of the run, were captured using the camera provided with the microscope. The *VB* for each run was worked out by taking average of the measurements taken on the four teeth. A Kistler piezoelectric dynamometer 9265 B, along with a force plate 9443 B, was used to measure the milling forces. The dynamometer can measure a maximum of 15 kN in each of the x- and y-directions and 30 kN in z-direction. The specific cutting energy was evaluated using three current clamp meters, Hantek CC 65, on the 3-phase input power supply of the CNC machining center. The total electrical power taken by the machine tool was calculated by the following relationship:*P*_total_ = 0.577(*I*_1_ + *I*_2_ + *I*_3_).*PF*.*V*(1)
where *I*_1_, *I*_2_, and *I*_3_ are the current values measured by the clamp meters, *V* is the voltage (measured as 220 V), and *PF* is the power factor (measured as 0.85). *P*_total_, evaluated for each run, is the summation of the cutting power and the noncutting power. To quantify the latter, the cutter was rotated and moved linearly, without contacting the work, at the four combinations of cutting speed and feed speed as presented in Table 1. The resulting noncutting powers are subtracted from the *P*_total_ to obtain the values of the cutting power. The cutting power for each run is divided by the corresponding *MRR* (shown in Table 1) to obtain the value of the specific cutting energy.

## 3. Results

This section presents the experimental results regarding tool wear, surface roughness, specific cutting energy, and milling forces.

### 3.1. Tool Wear

Figure 2 presents the experimental results regarding *VB*. The plots were categorized with respect to the three coolant’s options. Equal scales on the ordinates of the three plots were selected to have fair visual comparisons of the results. The plots show that there is a general tendency of a rise in tool wear with an increase in cutting speed. However, the difference lies in their slopes. The upsurge in wear is significantly steeper in dry milling in general, and in throttle cryogenic cooling employing the smallest coolant’s flow rate. In all the other cases, the coolants succeeded to nearly level off the wear growth with respect to the increase in cutting speed. This observation illustrates the potential of the fluids for enabling increased material removal rates and thus higher productivities without incurring higher wear rates and tooling costs. The reason behind the observation is the effective dissipation of process heat through absorption by the supercool fluids supplied in appropriate quantities, which leads to deceleration of the temperature-dependent tool damage modes. Because of this, the smallest flow rate of the throttle cryogenic coolant is not so effective in curbing tool wear at the higher cutting speeds due to its much higher working temperature than the evaporative coolant. It is reasonable that a not-so-cold fluid needs a higher quantity (or flow rate) to be equally effective. Therefore, much larger gaps are visible in the data points for the throttle coolant at *V*_c_ = 200 m/min as compared to those at the lower levels of cutting speed. For the evaporative coolant, a higher flow rate does improve tool wear conditions at all the levels of cutting speed, but the difference is not as significant.

The *VB* data at the lowest level of cutting speed is also worthy of comparison among the three coolant options. At *V*_c_ = 50 m/min, the most and the least effective wear control means are provided by the throttle and the evaporative cryogenic coolants, respectively. The latter’s underperformance is attributed to its extremely low working temperature, through which it also cools down the work material. The low cutting speed (50 m/min) causes the LN_2_ jet to impact the work surface, in addition to the tool, for a considerably long period of time, leading to reduction in its temperature. The resulting cooling of the work material raises its flow stress (instantaneous yield strength of a material being deformed), thus making it more difficult to cut.

Figure 3 presents the micrographs of the milling cutters used in the selected five runs. Progressive mechanical wear and adhesion of work material are the two prominent modes of tool damage visible from the images. The intensity of adhesion is clearly dependent on the cooling mode. Of the three options of coolant, dry milling (Figure 3a) accumulated the severest level of adhesion. The images also reveal that the intensity of adhesion also depends on the coolant’s mass flow rate. In comparing Figure 3b with Figure 3c and then Figure 3d with Figure 3e, we can observe that the intensity of adhesion is curbed when the coolant’s mass flow rate is increased. Furthermore, at the same levels of cutting speed and the coolant’s mass flow rate, the throttle cryogenic cooling yielded milder adhesion than the evaporative one. For all the cooling options and mass flow rates, the cutting speed was found to have a positive relationship with the intensity of adhesion, which is attributed to an increase in the rate of heat generation, causing higher adhesive tendency of the work material.

Progressive mechanical wear also has an uphill relationship with cutting speed for dry milling and milling under throttle cryogenic cooling with a small flow rate. The observation is attributed to the cooling adequacy of the applied coolant with respect to the rate of heat generated at the given level of cutting speed. The cooling potential of the coolant is dependent on its working temperature as well as mass flow rate. This is why the evaporative cryogenic coolant, being exceedingly cooler than the throttle one, can level off the *V*_c_-*VB* curve even at the smallest mass flow rate.

### 3.2. Specific Cutting Energy

SCE is an important sustainability measure as it indicates the amount of electrical energy consumed per unit volume of work material removed. More consumption of energy leads to more of its generation, causing depletion of earth resources and increased emissions of greenhouse gases.

Figure 4 presents the experimental results regarding SCE. A general trend that can be observed from the graphs is that an increase in cutting speed causes a diminution in SCE. The reason behind the observation is that the power required to remove the work material increases with a lower proportion as compared to the gain in material removal rate when the cutting speed is increased. With regard to the option of coolants, the evaporative cryogenic coolant and dry cutting come up as the worst performers at the lowest and highest cutting speeds, respectively. The former is attributed to the supercooling of the work material at a low cutting speed, leading to a hike in its flow stress. An increased flow stress demands higher consumption of energy for the material’s plastic deformation. Overall, the throttle cryogenic coolant yields the lowest values of SCE because of its effective heat dissipation mechanism and its not-so-cold effect on the work material.

Because of its extremely low temperature, the highest flow rate of the evaporative cryogenic coolant is the most suitable one to tackle the high rate of heat generation experienced at the highest level of cutting speed. Likewise, the lowest flow rate is appropriate for cooling at the lowest level of cutting speed because the rate of heat generation is low, and the work material needs to be protected from getting overcooled. The same reason is reflected by the *SCE* results related to the evaporative cryogenic cooling in Figure 4.

### 3.3. Work Surface Roughness

Figure 5 shows the experimental results regarding the averaged arithmetic surface roughness of the milled side walls. The plots show a general tendency of a fall in *R*_a_ as the cutting speed is increased from its lowest value to somewhere in between 100 and 150 m/min. From 150 to 200 m/min, the surface roughness value rises again except for the throttle cryogenic cooling, where the *R*_a_ holds steady. Generally, an increase in cutting speed is expected to generate better surface finish due to more mechanical and less thermal effect on the work material. However, this trend has a limit, given that a higher speed is accompanied with more heat entering the work and a higher wear rate that deteriorates the cutting edges, leading to a poorer surface finish. Of the 28 runs, the best surface finish (*R*_a_ = 0.82 µm) was produced by the highest mass flow rate of the throttle cryogenic coolant when applied at the cutting speed of 150 m/min.

Both of the cryogenic coolants largely yielded better surface quality than dry milling. A mutual comparison between the two coolants suggests that the throttle coolant yields a better surface finish at the lower levels of cutting speed whereas the evaporative coolant outperforms the others at the higher levels. The reason behind the observation is that a richer cooling potential of LN_2_ wards off the extra heat transferring into the work and avoids severe damage of the cutting edges at the higher levels of cutting speed.

Figure 6 shows the magnified views of the milled surfaces generated in the selected runs. A pattern of vertical stripes (symbolic of an interrupted machining process) is common in all the six images. Depth of these stripes has a direct impact on the measure of *R*_a_. Three of the images (Figure 6a,b,e) also show adhesions of work material chips on the surfaces. These adhesions, which contribute to deterioration of the machined surfaces, are observed in all the runs of dry milling and the runs involving throttle cryogenic cooling carried out at the lowest level of mass flow rate. This suggests that a cryogenic coolant also avoids adhesions of chips/microchips on the milled surface when it is operated at a very low temperature or applied in an adequate quantity.

### 3.4. Machining Forces

Figure 7 presents the experimental results for *F*_xy_ and *F*_z_. For dry milling, linear increases in the magnitudes of both of the components in correlation with cutting speed are visible, although the magnitude of *F*_z_ is generally less than half of *F*_xy_. Of the 28 runs, the largest magnitudes of the force components were observed in dry milling carried out at the highest level of cutting speed.

The plots suggest that the cryogenic coolants do have a restraining effect on the milling forces. In particular, the evaporative coolant, due to its extremely low working temperature, tends to keep the plots nearly horizontal. The throttle coolant yields lower magnitudes of the forces than the evaporative one at the lower levels of cutting speed. This is so because the not-so-cold coolant does not tend to strengthen the work material at low cutting speeds (and feed speeds). On the other hand, the throttle coolant loses a bit of its cooling potential at the higher cutting speeds and thus need higher flow rates to cope with the increased heat generation rates.

### 3.5. Specific Fluid Consumption

*SFC* is a novel sustainability measure that quantifies the quantity of a cryogenic coolant consumed per unit volume of work material removed. As the axial and radial depths of cut were fixed for all the runs, “per unit length of cut” can replace “per unit volume removed” in the measure. Thus, SFC, in this work, is the mass of a cryogenic fluid consumed in cutting a unit length of the work. As described earlier, it is obtained by dividing the given mass flow rate of a coolant with the tool’s feed speed. Table 2 presents the SFCs for different mass flow rate—cutting speed combinations used in this work. A low value of this measure is desired for sustainability. The lowest value (0.063 kg/m) was yielded when the mass flow rate and cutting speed were set to 0.2 kg/min and 200 m/min, respectively. Although a low value of SFC conserves a vital resource and keeps a check on process cost, it does not yield favorable values of the other sustainability measures, as is detailed in the preceding subsections.

## 4. Evaluation of Machining Sustainability

Sustainability stands on three pillars: social, environmental, and economic. By assuming that the two cryogenic coolants impose negligible health hazards and safety concerns when utilized in a controlled environment for heat dissipation in machining, as detailed in [3], the sustainability comparison, across the parameters, can be performed by discarding the social aspect. The method devised in this work to quantify sustainability for each run is based on assigning weights to various economic and environmental measures of the milling process. It is outlined as follows:The sustainability score (SS) is a real number between 0 and 100. The higher the SS of a run (or combination of the control parameters), the better it is collectively for the environment and the economy of the process.The SS is contributed by the following factors possessing the weightages shown in the parentheses: (a) specific tool damage (35), which is quantified as the magnitude of tool wear incurred against a given volume of material removed, hence, VB; (b) productivity (25), which is quantified through MRR; (c) the coolant’s consumption cost (17); (d) specific cutting energy (15); and (e) the product quality (7), which is quantified, in this work, in terms of R_a_. The weightages of the five factors sum to 100. It should be noted that most of the aforementioned factors collectively hold economic as well as environmental values. For instance, specific tool damage not only governs the tooling cost (as a higher wear rate causes more frequent tool replacements) but also affects resource consumption, landfill, and tool’s embodied energy. The weightages assigned to the factors are based on their desirability levels as usually sought in the machining industry. Nevertheless, the values are mutable and can be adjusted according to the priority mix of the objectives on hand.Specific coolant’s consumption cost (SC3), in CNY/m, is calculated by multiplying the coolant’s acquisition cost (12 CNY/kg and 3 CNY/kg, respectively, for the evaporative and throttle cryogenic coolants) with the respective SFC (kg/m) of the run. It can be seen that the SC3 with respect to the runs involving dry milling is zero. Figure 8 presents the SC3 values with respect to the two cryogenic coolants. Clearly, of the two, the throttle cryogenic cooling is the more economical option.Finally, the SS for each run is calculated by using the following formula: (2)$$SS=35\left(\frac{VB_{max}-VB}{VB_{max}-VB_{min}}\right)+25\left(\frac{MRR-MRR_{min}}{MRR_{max}-MRR_{min}}\right)+17\left(\frac{SC3_{max}-SC3}{SC3_{max}-SC3_{min}}\right)\;+15\left(\frac{SCE_{max}-SCE}{SCE_{max}-SCE_{min}}\right)+8\left(\frac{R_{a}{}_{max}-R_{a}}{R_{a}{}_{max}-R_{a}{}_{min}}\right)$$

The multipliers 35, 25, 17, 15, and 8 are the weightages of the contributing factors as described in Step 2. The maximum and minimum values of the parameters used in the equation are the corresponding maximum and minimum values of the respective data presented in this work (Figure 2, Figure 4, Figure 5 and Figure 8 and Table 1).

The SS values for all the runs obtained by applying the formula presented in Equation (2) are shown in Table 3. The rows of the table are arranged in the descending order of SS to have a better view of the influential parameters and their favorable levels.

## 5. Discussion

A few important conclusions can be drawn from Table 3. The top slots, with respect to the SS, are occupied by the runs employing high cutting speeds and high flow rates of the two cryogenic coolants. This means that the high consumption rates of the cryogenic fluids paid the dividend by enabling high cutting speeds and hence high MRRs (productivity), without being impaired significantly by high tool wear rates and the resulting tooling costs. An increased difference in the magnitude of tool wear with an increase in cutting speed between a cryogenic and noncryogenic cooling condition was also reported in another study [21]. The highest sustainability score (91.4) is seen in the run involving the throttle coolant and the highest levels of cutting speed and mass flow rates. Its score is about 7 points more than the second best, whereas the differences of the scores between the second and the third and between the third and the fourth are both less than 1. This suggests that the combinations of the throttle coolant and the highest levels of cutting speed and coolant’s mass flow rate is the obvious choice for the sustainable milling of Ti–6Al–4V. Of the first 14 positions with respect to *SS*, the throttle and evaporative cryogenic coolants have taken 9 and 5 of them, respectively. This shows that throttle cryogenic cooling using compressed CO_2_ gas offers a more sustainable option of milling titanium alloy than LN_2_ does. The higher standing of the former comes not only because of its effective heat dissipation mechanism but also due to its low acquisition cost, which matters more when a coolant is consumed in a high quantity. Moreover, it is quite surprising to note that the last positions in the table are not taken by the runs involving dry milling but by the ones employing evaporative cryogenic cooling and the lowest levels of cutting speed. The effect of work strengthening due to the impingement of supercool LN_2_ on the work at a slow feed speed renders the cooling method ineffective in controlling the tool wear. Khanna and Agrawal also reported a significant growth in power consumption when the mode of titanium machining is changed from dry to cryogenically cooled at a low level of cutting speed [22]. The ineptness in checking tool wear coupled with its high acquisition cost makes the evaporative cryogenic coolant highly unsustainable at the lowest level of cutting speed. To corroborate, Bhat et al. also showed that the sustainability level of cryogenic machining overtakes that of dry machining only at high cutting conditions [23].

Dry cutting has shown its serious limitation regarding sustainability of titanium milling. Fluidless cutting remains highly unsustainable irrespective of the cutting speed employed. The highest *SS* it could muster is just 61.5, which happened at a medium level of cutting speed. A decrease in cutting speed would further cutback the *SS* because of the reduction in productivity and the hike in specific cutting energy. On the other hand, an increment to the highest level of cutting speed would also cause a drop in *SS* because of the massive growth in tool wear rate and degradation of work surface quality. On average, a 40% improvement in work surface finish was reported due to the application of a cryogenic fluid [22].

With respect to flow rate/cutting speed trade-off, it should be noted that a high level of cutting speed is desired whereas that of a coolant’s mass flow rate is not. In this context, the combination of highest cutting speed and lowest flow rate sits at the seventh position (SS = 75.9) for the evaporative coolant and eleventh (SS = 71) for the throttle one. Although the situation is acceptable, considering the median *SS* of 66.5 for the 28 runs, the increments in flow rate from 0.2 to 0.4 kg/min and from 0.4 to 0.6 kg/min resulted in the enhancement in sustainability for both of the cryogenic coolants. Specifically, the increments in the case of the throttle coolant are larger than the other. Thus, it can be concluded that the milling of Ti–6Al–4V should be carried out at the high levels of cutting speed, and the heat dissipation through a cryogenic coolant should be done at high mass flow rates.

## 6. Conclusions

The presented work analyzed and quantified the sustainability of milling Ti–6Al–4V under cryogenic conditions against the different combinations of coolant’s mass flow rate and cutting speed. The sustainability quantification was performed from the economic and environmental perspectives. The following conclusive points can be drawn from the experimental results and analyses provided above:A high cutting speed and the application of a throttle cryogenic coolant with a high mass flow rate can yield the highest level of sustainability in the milling of titanium alloy. Such a favorable outcome is attributed to the high productivity, adequate cooling capacity for tool wear abatement, low energy consumption, and acceptable work surface quality.The evaporative cryogenic coolant also weighs in its highest sustainability potential when it is applied with a high mass flow rate and against a high cutting speed of milling. However, its performance marginally falls behind that of the throttle cryogenic coolant because of its high acquisition cost and exceedingly low working temperature that leads to an increase in specific cutting energy consumption.Dry milling of the titanium alloy is highly unsustainable, especially when the cutting is done at a very high or a very low level of cutting speed. The high and low levels of cutting speed render the process unsustainable due to an exceedingly high tool wear rate and unacceptably low productivity and the associated high specific energy consumption, respectively.

## Figures and Tables

**Figure 1 materials-14-03429-f001:**
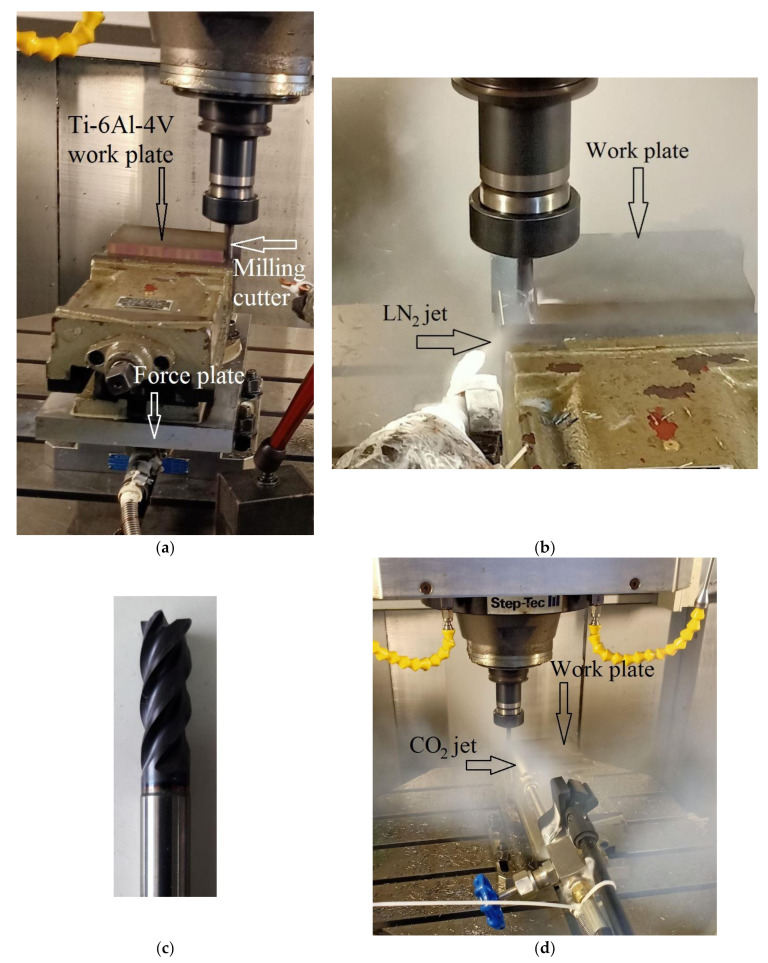
The experimental setup: (**a**) the work plate, cutter, and force plate; (**b**) milling under evaporative cryogenic cooling; (**c**) the side- and end-mill cutter used in the experiments; and (**d**) milling under throttle cryogenic cooling.

**Figure 2 materials-14-03429-f002:**
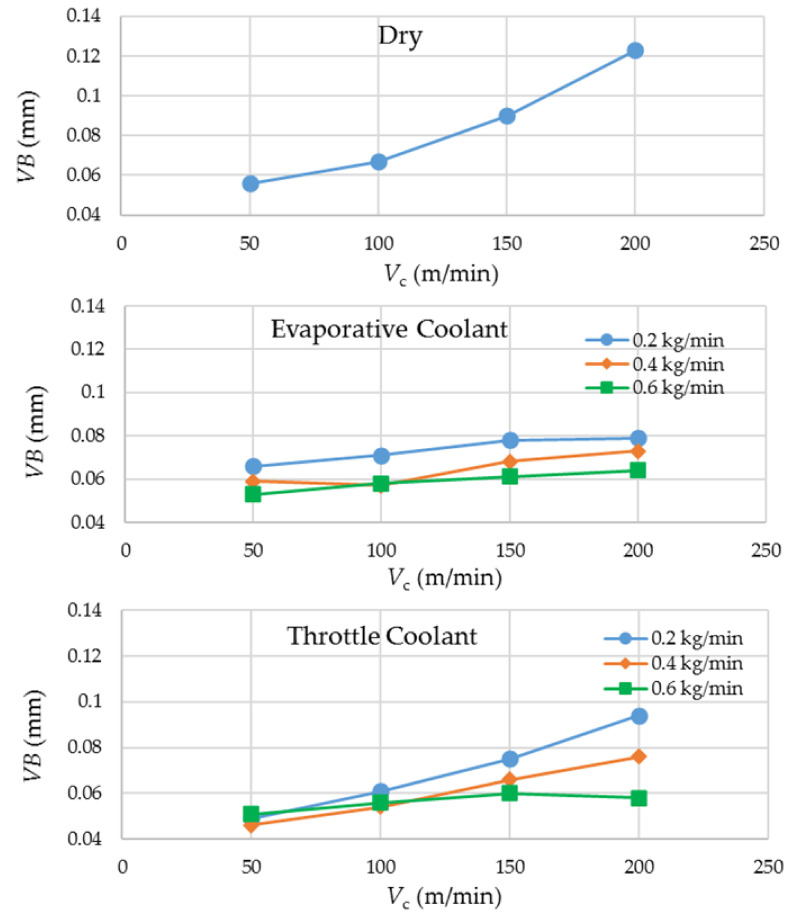
Experimental results regarding tool wear, categorized with respect to the choice of coolant.

**Figure 3 materials-14-03429-f003:**
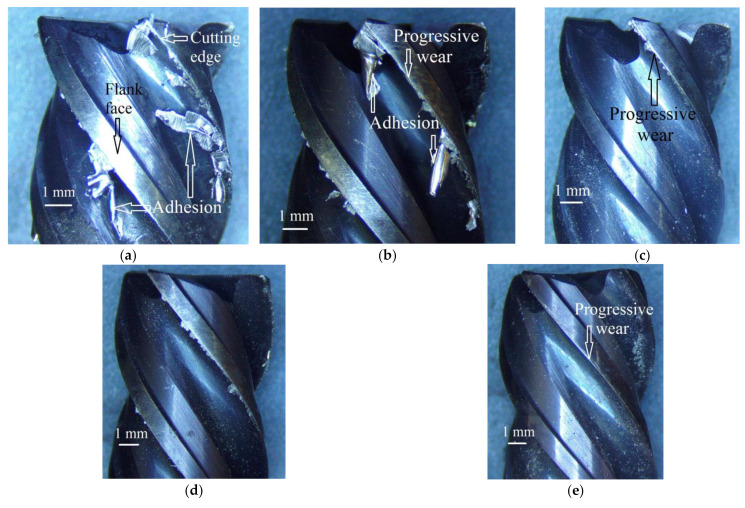
Micrographs of the milling cutters used in the following runs: (**a**) coolant = dry, *V*_c_ = 150 m/min; (**b**) coolant = evaporative, flow rate = 0.2 kg/min, *V*_c_ = 200 m/min; (**c**) coolant = evaporative, flow rate = 0.6 kg/min, *V*_c_ = 200 m/min; (**d**) coolant = throttle, flow rate = 0.4 kg/min, *V*_c_ = 200 m/min; and (**e**) coolant = throttle, flow rate = 0.6 kg/min, *V*_c_ = 200 m/min.

**Figure 4 materials-14-03429-f004:**
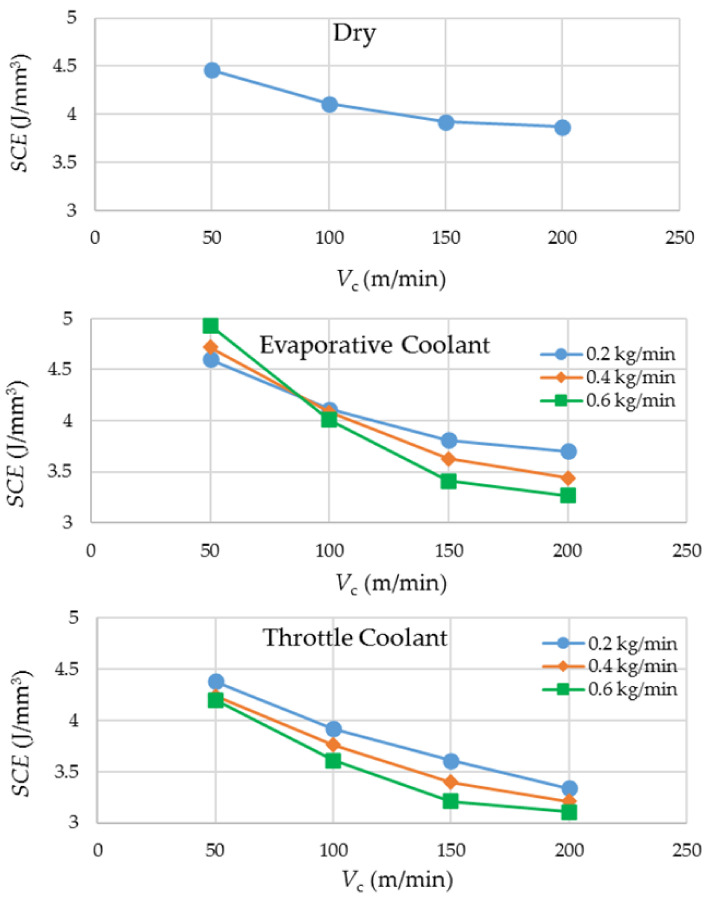
Experimental results regarding specific cutting energy.

**Figure 5 materials-14-03429-f005:**
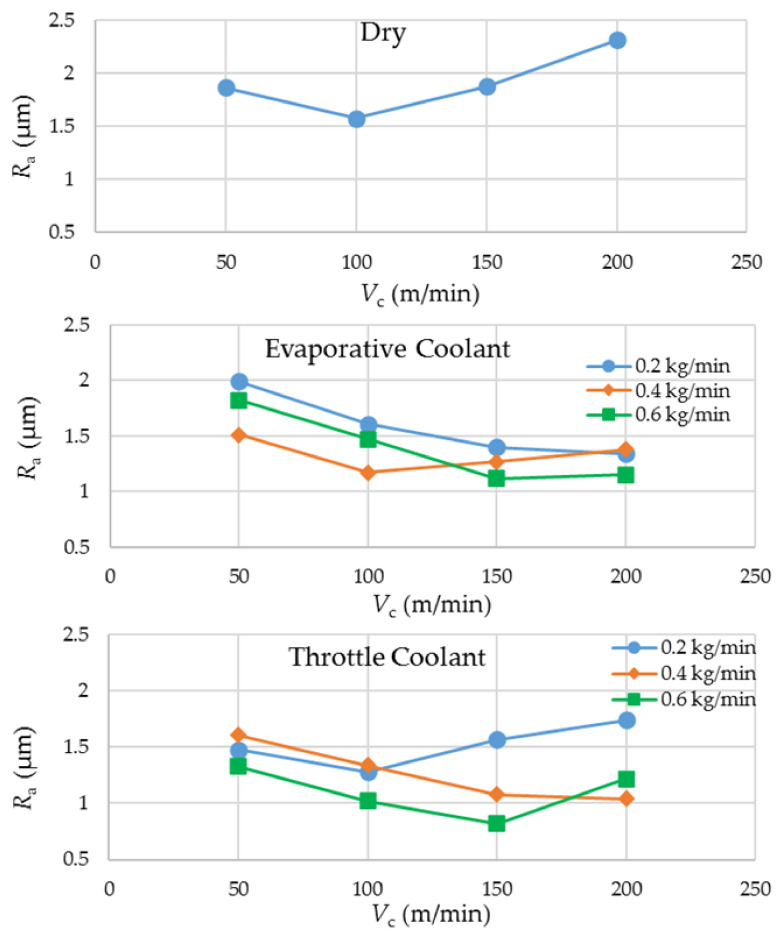
Experimental results regarding work surface roughness.

**Figure 6 materials-14-03429-f006:**
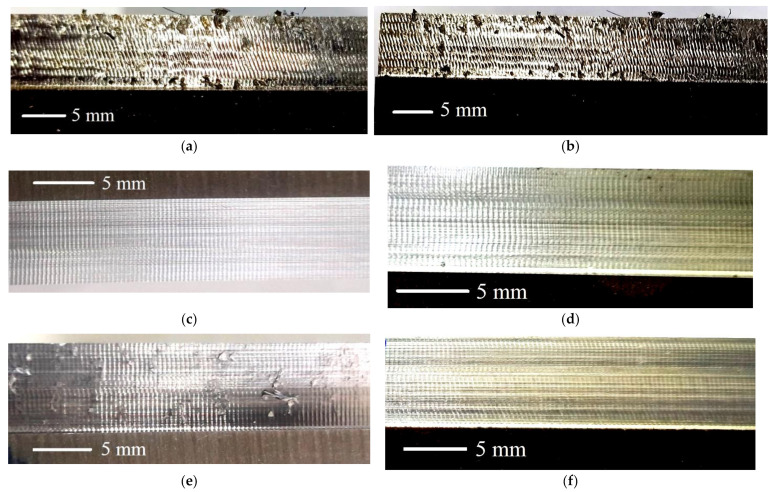
Magnified views of the milled surfaces generated in the following runs: (**a**) coolant = dry, *V*_c_ = 50 m/min; (**b**) coolant = evaporative, flow rate = 0.2 kg/min, *V*_c_ = 50 m/min; (**c**) coolant = evaporative, flow rate = 0.4 kg/min, *V*_c_ = 150 m/min; (**d**) coolant = evaporative, flow rate = 0.6 kg/min, *V*_c_ = 200 m/min; (**e**) coolant = throttle, flow rate = 0.2 kg/min, *V*_c_ = 50 m/min; and (**f**) coolant = throttle, flow rate = 0.6 kg/min, *V*_c_ = 200 m/min.

**Figure 7 materials-14-03429-f007:**
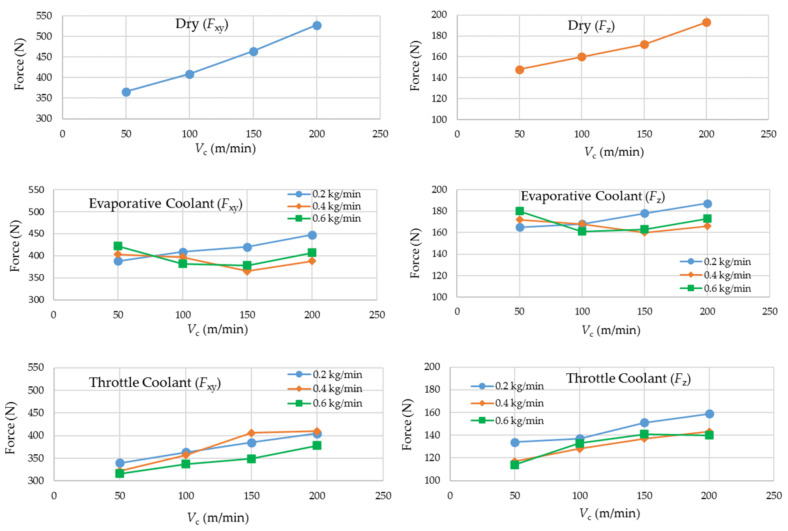
Experimental results regarding *F*_xy_ (resultant of the coplanar components) and *F*_z_ (the component acting along the tool’s axis).

**Figure 8 materials-14-03429-f008:**
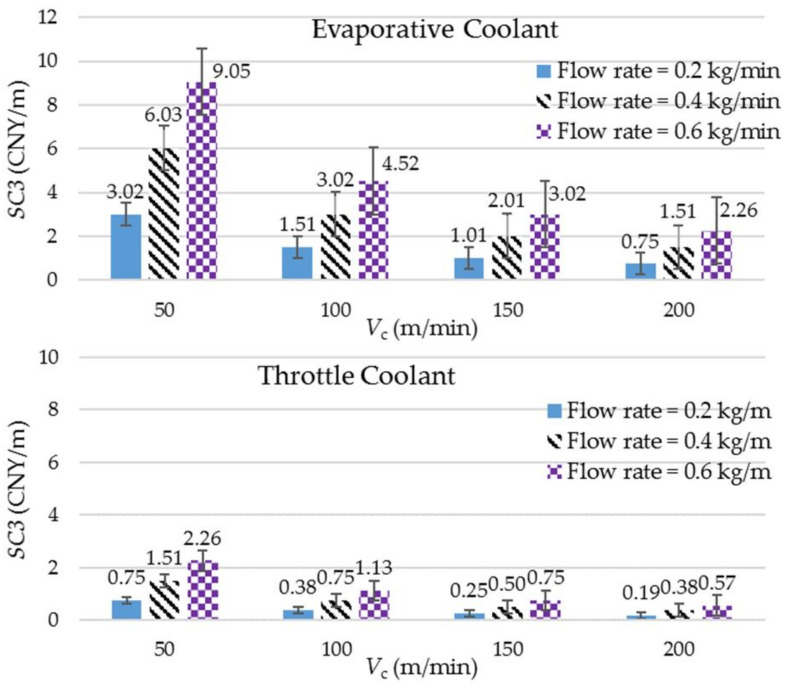
The calculated values of specific coolant consumption cost against different combinations of cutting speed and coolant’s mass flow rate.

**Table 1 materials-14-03429-t001:** The values of feed speed (*V*_f_) and material removal rate (*MRR*) against the four levels of cutting speed (*V*_c_) and feed rate of 0.1 mm/tooth.

*V*_c_ (m/min)	*V*_f_ (mm/min)	*MRR* (mm^3^/s)
50	795.8	46.4
100	1591.6	92.8
150	2387.3	139.3
200	3183.1	185.7

**Table 2 materials-14-03429-t002:** The values of specific fluid consumption at various combinations of coolant’s mass flow rate and cutting speed.

Flow Rate (kg/min)	V_c_ (m/min)	SFC (kg/m)	Flow Rate (kg/min)	V_c_ (m/min)	SFC (kg/m)	Flow Rate (kg/min)	V_c_ (m/min)	SFC (kg/m)
0.2	50	0.251	0.4	50	0.503	0.6	50	0.754
	100	0.126		100	0.251		100	0.377
	150	0.084		150	0.168		150	0.251
	200	0.063		200	0.126		200	0.188

**Table 3 materials-14-03429-t003:** Sustainability scores (SS) for the 28 runs, with rows arranged in the descending order of the SS values.

S/No.	Coolant	Flow Rate (kg/min)	*V*_c_ (m/min)	SS	S/No.	Coolant	Flow Rate (kg/min)	*V*_c_ (m/min)	SS
1	Throttle	0.6	200	91.4	15	Evaporative	0.2	150	66.3
2	Evaporative	0.6	200	84.5	16	Evaporative	0.4	100	62.8
3	Throttle	0.4	200	83.7	17	Dry	-	100	61.5
4	Throttle	0.6	150	83.0	18	Dry	-	150	59.3
5	Evaporative	0.4	200	79.2	19	Throttle	0.4	50	58.6
6	Throttle	0.4	150	77.8	20	Evaporative	0.6	100	58.5
7	Evaporative	0.2	200	75.9	21	Throttle	0.2	50	58.2
8	Evaporative	0.6	150	75.1	22	Throttle	0.6	50	56.8
9	Throttle	0.6	100	71.5	23	Evaporative	0.2	100	56.7
10	Evaporative	0.4	150	71.2	24	Dry	-	50	53.7
11	Throttle	0.2	200	71.0	25	Dry	-	200	50.7
12	Throttle	0.4	100	70.1	26	Evaporative	0.2	50	41.7
13	Throttle	0.2	150	69.9	27	Evaporative	0.4	50	40.8
14	Throttle	0.2	100	66.7	28	Evaporative	0.6	50	34.5

## Data Availability

The data presented in this study are available on request from the corresponding author. The data are not publicly available due to the possibility of furtherance of research covering different aspects of manufacturing sustainability.

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
