# Peer review of "On Coolant Flow Rate-Cutting Speed Trade-Off for Sustainability in Cryogenic Milling of Ti–6Al–4V"

_materials, 2021, doi:10.3390/ma14123429_

Round 1
Reviewer 1 Report
The authors present a very interesting study in determining the optimum cooling parameters to achieve a sustainable machining process of titanium alloy. The context, importance of the study and objective are well described and defined in the Introduction. The Materials and Methods section provides sufficient details of materials and experimental protocols allowing other to reproduce the work without difficulty. The results are well presented and interpreted that lead to a meaningful discussion bringing new information and understanding to the field of titanium machining.
Few additional comments:
- The images can be better organized, the tables can be better presented by not just using a plain basic feature of Excel.
- Speaking about CO2 gas cooling, is there a system in use to capture the gas so it will not pollute the environment?
- It would be more helpful for the readers if the authors add a Conclusion section to organize their conclusive remarks, the gained new understanding for the advancement of knowledge in titanium machining, and the take home messages for the readers.
Author Response
Reviewer 1: (The changes done in the manuscript are shown in BLUE)
Comment # 1:
The authors present a very interesting study in determining the optimum cooling parameters to achieve a sustainable machining process of titanium alloy. The context, importance of the study and objective are well described and defined in the Introduction. The Materials and Methods section provides sufficient details of materials and experimental protocols allowing other to reproduce the work without difficulty. The results are well presented and interpreted that lead to a meaningful discussion bringing new information and understanding to the field of titanium machining.
Response:
Thank you very much for your appreciation.
Comment # 2:
The images can be better organized, the tables can be better presented by not just using a plain basic feature of Excel.
Response:
All the three tables are prepared by directly using the template provided by MDPI / Materials’ Editorial Office. All the fine details of the tables, including font size, alignment, border colors, etc., are in accordance with the template.
The presentation of the graphs (Figures 2, 4, 5, and 8), coming from Excel, has now been improved.
Comment # 3:
Speaking about CO2 gas cooling, is there a system in use to capture the gas so it will not pollute the environment?
Response:
Unfortunately, it is quite difficult to re-capture the gas because it starts mixing with the air as soon as it starts throttling out of the nozzle. It becomes as difficult to capture the used CO2 gas as is to distill it from the atmospheric air. Capturing the gas from the air would surely neutralize the effect of compressed CO2 cooling but the commercially available processes are highly inefficient and expensive. Thus, the gas is commonly obtained by burning waste material and fossil fuels. Nevertheless, the quantities used for cooling in the machining processes are not so high to cause a significant increase in CO2 footprint.
Comment # 4:
It would be more helpful for the readers if the authors add a Conclusion section to organize their conclusive remarks, the gained a new understanding for the advancement of knowledge in titanium machining, and the take home messages for the readers.
Response:
“Conclusions” section has now been added after the “Discussion” section.
Reviewer 2 Report
The authors have prepared an interesting article in the field of manufacturing engineering. The experiment is described correctly. The elaboration of the results is understandable but not all aspects of the results are explained. The discussion is insufficient. There is no reference to other works on similar topics and no critical analysis. The reviewer suggests improving the work in this regard.
Author Response
Reviewer 2: (The changes done in the manuscript are shown in GREEN)
Comment:
The authors have prepared an interesting article in the field of manufacturing engineering. The experiment is described correctly. The elaboration of the results is understandable but not all aspects of the results are explained. The discussion is insufficient. There is no reference to other works on similar topics and no critical analysis. The reviewer suggests improving the work in this regard.
Response:
Thank you very much for the suggestion. The Discussion section has now been expanded and references to other works, on same topics, are included in it.
Reviewer 3 Report
The authors are presenting a paper on milling of titanium. The experimental work is adequate and it can be considered as comprehensive. I would like to make a few comments:
1) Use SI for units. For instance, use l instead of L, s instead of sec.
2) In figures 4 and 5, for instance, there are several crossings between lines. What is happening in here? Could it be related to errors in measuring or during the machining process?
3) Regarding the whole concept. I'm not sure about your sustainability index. For instance, tool wear and material removal rate are mainly related to productivity. For sure, the need of new tools has an impact but you are not addressing it. This impact depends mainly on the type of material, the tool wear criteria for replacing that may depend on the tool material. Moreover, What is the influence of the surface quality on the sustainability? In my opinion there is no relation. And also related to the cooling solutions. What is the relation of the cost of the solution to the sustainability? None. If a solution is sustainable, the cost is not relevant. In fact, if you are using CO2 for instance, you are working with CO2 and LN2, so you should answer the question, are these materials sustainable? In any case, you can consider the requirements in terms of enegy for pumping the coolant and that would be related to the flow rate, not to the cost in my opinion.
Author Response
Reviewer 3: (The changes done in the manuscript are shown in DARK BLUE)
Comment # 1:
The authors are presenting a paper on milling of titanium. The experimental work is adequate and it can be considered as comprehensive. I would like to make a few comments:
Use SI for units. For instance, use l instead of L, s instead of sec.
Response:
Thank you very much. The needful is done.
Comment # 2:
In figures 4 and 5, for instance, there are several crossings between lines. What is happening in here? Could it be related to errors in measuring or during the machining process?
Response:
No. Absolutely not. There is no process or measurement error involved. Just, some of the measured responses are changing more intensely with cutting speed at one level of coolant’s mass flow rate than at the other. That’s why the data lines are seen crossing others. Agreeably, the crisscrossing is more prominent in Ra. The complex nature of its dependence on the control parameters makes this erratic observation possible.
Comment # 3:
Regarding the whole concept. I'm not sure about your sustainability index. For instance, tool wear and material removal rate are mainly related to productivity. For sure, the need of new tools has an impact but you are not addressing it. This impact depends mainly on the type of material, the tool wear criteria for replacing that may depend on the tool material. Moreover, What is the influence of the surface quality on the sustainability? In my opinion there is no relation. And also related to the cooling solutions. What is the relation of the cost of the solution to the sustainability? None. If a solution is sustainable, the cost is not relevant. In fact, if you are using CO2 for instance, you are working with CO2 and LN2, so you should answer the question, are these materials sustainable? In any case, you can consider the requirements in terms of energy for pumping the coolant and that would be related to the flow rate, not to the cost in my opinion.
Response:
We would like to address this comment in segments:
For instance, tool wear and material removal rate are mainly related to productivity => Material removal rate is surely related to productivity, but tool wear is not, at least, in a direct manner. It is more related to processing cost. The higher the tool wear rate, the quicker the tool life criterion is met, and the tool replaced and, thus, the higher is the processing cost. It is also related to landfilling, an environmental factor. More frequent tool replacements result in a higher level of landfilling.
For sure, the need of new tools has an impact but you are not addressing it => We have surely addressed this impact in our calculations. Please see lines 406 – 408. Also, in the formula for SS (equation 2), the term VBmax is the tool life criterion (as is famously known). When the VB reaches VBmax, the tool is supposed to be replaced.
This impact depends mainly on the type of material, the tool wear criteria for replacing that may depend on the tool material => Tool wear criterion is just discussed above. Regarding the type of work material, the presented work is related to the machining of Ti-6Al-4V only. The title of the article also specifies the scope in this regard.
Moreover, What is the influence of the surface quality on the sustainability? In my opinion there is no relation => Surface quality does have a direct influence on sustainability. A poor quality may cause part’s rejection in quality inspection, which may have 2 consequences: (1) rework of the part, leading to more consumption of energy and cutback in productivity; and (2) even worse, scrapping of the part, leading to increase in production cost and intensification of landfilling or recycling.
And also related to the cooling solutions. What is the relation of the cost of the solution to the sustainability? None. If a solution is sustainable, the cost is not relevant => We do not agree with this. Economy (hereby quantified in terms of cost) is one of the three pillars of sustainability. How can a solution be sustainable if it is not cost effective?
In fact, if you are using CO2 for instance, you are working with CO2 and LN2, so you should answer the question, are these materials sustainable? In any case, you can consider the requirements in terms of energy for pumping the coolant and that would be related to the flow rate, not to the cost in my opinion => N2 is a harmless gas, abundantly available in the atmosphere. CO2 gas does not pose any asphyxiation threat in the flow rates we have tested in the work. Moreover, the minute quantity of it used for heat dissipation purposes does not increase the carbon footprint. Furthermore, the transportation of these gases from dewar/bottle to the cutting areas does not require any external source of power/energy.
Reviewer 4 Report
The authors of the paper : On Coolant Flow Rate-Cutting Speed Tradeoff for Sustainability in Cryogenic Milling of Ti-6Al-4V present some important experimental results in the field of biomedical applications. Few aspects must be clarified before publication:
Introduction is too big and must be better synthesized
L206: Figure 1: image c) can be used as detail for b)
L322: how many determinations of roughness were made ?
L339: Fig 6 : please insert a scale for a)-f) images in mm or cm
L428: can you provide in figure 8 a Standard deviation of the values
A major concern: Please insert a section 6 Conclusions with main conclusions of the study after Section 5 Discussion
Author Response
Reviewer 4: (The changes done in the manuscript are shown in PURPLE)
Comment # 1:
The authors of the paper : On Coolant Flow Rate-Cutting Speed Tradeoff for Sustainability in Cryogenic Milling of Ti-6Al-4V present some important experimental results in the field of biomedical applications. Few aspects must be clarified before publication:
Introduction is too big and must be better synthesized
Response:
Thank you very much for your appreciation. The introduction section is reduced in size (from 1,140 words to 1,054 words). Its synthesis has also been improved.
Comment # 2:
L206: Figure 1: image c) can be used as detail for b)
Response:
Although not incorrect, we believe it may not be appropriate as, in this way, image (c) is not only the detail of image (b) but of images (a) and (d) as well.
Comment # 3:
L322: how many determinations of roughness were made ?
Response:
This detail is provided in Section 2.4 (L206 – L209): “Four measurements are taken at the distances of 20, 40, 60, and 80 mm from the starting point. The sampling length for each measurement is 4 mm. The Ra for each run is then evaluated by taking the average of the four measurements.”
Comment # 4:
L339: Fig 6 : please insert a scale for a)-f) images in mm or cm
Response:
All the six images have now been scaled in mm.
Comment # 5:
L428: can you provide in figure 8 a Standard deviation of the values
Response:
The needful is done. Error bars, representing the standard deviation of the data, are added in Fig. 8.
Comment # 6:
A major concern: Please insert a section 6 Conclusions with main conclusions of the study after Section 5 Discussion
Response:
The needful is done. The Conclusions section is added as the sixth section in the manuscript.
Round 2
Reviewer 4 Report
Publish as it is.